# Maladaptive Rumination Mediates the Relationship between Self-Esteem, Perfectionism, and Work Addiction: A Largescale Survey Study

**DOI:** 10.3390/ijerph17197332

**Published:** 2020-10-08

**Authors:** Bernadette Kun, Róbert Urbán, Beáta Bőthe, Mark D. Griffiths, Zsolt Demetrovics, Gyöngyi Kökönyei

**Affiliations:** 1Institute of Psychology, ELTE Eötvös Loránd University, H-1064 Budapest, Hungary; urban.robert@ppk.elte.hu (R.U.); demetrovics.zsolt@ppk.elte.hu (Z.D.); kokonyei.gyongyi@ppk.elte.hu (G.K.); 2Département de Psychologie, Université de Montréal, Montréal H2V 2S9, Canada; beabothe@gmail.com; 3Psychology Department, Nottingham Trent University, Nottingham NG1 4FQ, UK; mark.griffiths@ntu.ac.uk; 4NAP2-SE, Genetic Brain Imaging Migraine Research Group, Hungarian Academy of Sciences, Semmelweis University, H-1089 Budapest, Hungary; 5Department of Pharmacodynamics, Semmelweis University, H-1089 Budapest, Hungary

**Keywords:** work addiction, workaholism, self-esteem, perfectionism, rumination

## Abstract

*Background*: Empirical evidence suggests that low self-esteem and high perfectionism are significant personality correlates of work addiction, but the mechanisms underlying these relationships are still unclear. Consequently, exploring cognitive mechanisms will help to better understand work addiction. For instance, rumination is one of the under-researched topics in work addiction, although it may explain specific thinking processes of work-addicted individuals. The purpose of the study was to test the mediating role of maladaptive rumination (i.e., brooding) in the relationship between personality and addiction. *Methods*: In a largescale cross-sectional, unrepresentative, online study, 4340 adults with a current job participated. The following psychometric instruments were used: Work Addiction Risk Test Revised, Rosenberg Self-Esteem Scale, Multidimensional Perfectionism Scale, and Ruminative Response Scale. *Results*: It was found that self-oriented perfectionism, socially prescribed perfectionism, and self-esteem had both direct and indirect relationships with work addiction via the mediating effect of maladaptive rumination. The two paths involving brooding explained 44% of the direct relationship. *Conclusions*: The study demonstrated that brooding type of rumination as a putatively maladaptive strategy explains why individuals characterized by low self-esteem and high perfectionism may have a higher risk of work addiction. The results suggest that cognitive-affective mechanisms in work addiction are similar to those found in other addictive disorders.

## 1. Introduction

Work addiction is a ‘double-edged’ and controversial phenomenon, and several misbeliefs and myths have appeared in the literature concerning the disorder [1]. On one hand, work addiction can be easily regarded as a positive addiction [2], or as a useful behavioral pattern for employers and companies. Some individuals with work addiction have an elevated motivation to work including all the factors that commonly motivate people to work such as financial rewards, fidelity, and self-improvement [3]. On the other hand, work addiction and related overwork can have several adverse physical, mental, and social consequences both for individuals and their environment [4,5]. Although the prevalence of work addiction varies across surveys due to the different definitions, methods, and screening instruments used, largescale representative studies have shown that the lifetime prevalence of work addiction was 8.3% among Norwegian workers [6], and 8–9% among Hungarian employees [7,8]. Due to its seemingly high prevalence compared to other addictive behaviors [9], there is an increasing interest in understanding the psychosocial correlates of work addiction [4,10]. As far as the authors’ are aware, there are only two national representative studies investigating the prevalence of work addiction (i.e., [6,8]). However, societal and cultural factors might also contribute to the prevalence of the problem. For instance, Ng, Sorensen, and Feldman [11] posited that several socio-cultural factors may play important roles in the development of work addiction such as dysfunctional family experiences, vicarious learning at home or at work, peer competition at work, and work-related self-efficacy. Moreover, organizational culture can also intensify work addiction. For example, some organizations and industries reinforce excessive working or competitiveness. Consequently, the prevalence of work addiction might be elevated among employees [12]. Although the nature of work addiction can be explored by examining micro-, meso-, and macro-level characteristics [13], most studies have only focused on the individual factors of work addiction. Some research focusing on the personality correlates of work addiction concluded that the main personality traits (i.e., Big Five traits) play only a small part in the etiology of the disorder [14]. However, cognitive personality traits and processes may help to clarify the role of individual differences in work addiction. Consequently, the present study focuses on the relationship between personality and a specific cognitive process (i.e., rumination) in the etiology of work addiction.

### 1.1. Work Addiction

Work addiction was first defined by Oates [15,16], who described a “workaholic” as a “person whose need for work has become so excessive that it creates noticeable disturbance or interference with his bodily health, personal happiness, and interpersonal relations, and with his smooth social functioning” [16] (p. 7). Obsessiveness, as a key personality trait in work addiction, has also been emphasized in several approaches [2,17,18,19,20]. Porter [21] highlighted the cognitive mechanisms underlying work addiction including rigid thinking, obstinateness, and perfectionist attitudes. In the model posited by Schaufeli, Taris, and Bakker [22], the cognitive component of the construct (i.e., working compulsively) represents individuals frequently and persistently thinking about work, even if they are not in the workplace. In the cognitive-behavioral model of work addiction posited by Wojdylo et al. [23], dysfunctional cognitions, and affective and behavioral components are key components of work addiction, but the main factor is the affective or hedonic one: ‘work craving’ originates from the unrealistic standards of perfectionism. Consequently, in most of the theoretical models, work addiction is unequivocally based on personality factors. In addition to obsessive-compulsiveness, self-esteem and perfectionism have also been frequently mentioned as typical personality factors of work addiction.

### 1.2. Work Addiction, Perfectionism, and Self-Esteem

In the earliest theories of work addiction, perfectionism and self-esteem were identified as important antecedents. However, these theories were based on observations and clinical anecdotes [2,24,25,26,27] and the need for empirical studies became essential. Perfectionism—a personality dimension characterized by individuals setting very high norms of performance for themselves and being overcritical of their own behavior [28]—has been examined in relation to work addiction in a few studies and almost all of them have focused on the multidimensional nature of the construct. Among the different dimensions, ‘discrepancy’ (between one’s performance expectations and self-evaluations of current performance) and ‘concern over mistakes’ have been found to be among the most relevant correlates of work addiction [29,30]. Hewitt and Flett [31] distinguished between socially prescribed perfectionism (SPP), other-oriented perfectionism (OOP), and self-oriented perfectionism (SOP). SOP refers to individuals establishing extremely high standards for themselves, and the importance of being perfect and avoiding any failure. OOP refers to the expectation of unrealistic standards for significant others and evaluating others by the individual’s own high standards. Finally, SPP contains the individual’s beliefs about others’ high expectations of themselves, and beliefs about being accepted by others only in the case of reaching these standards and being perfect for them.

According to early theories of work addiction [18,24], parents of individuals with this problem set very high standards for their children, resulting in low self-esteem and high perfectionism. If the child always has to be ‘good’ and behave as an adult, it is predicted that their self-oriented perfectionism and socially prescribed perfectionism will be higher than others. Stoeber, Davis, and Townley [32] found that among the three perfectionism components, only SOP was a significant positive predictor of work addiction; interestingly, SPP was unrelated to work addiction. However, this study comprised a very small sample (*N* = 131) with only two types of participants (caravan owners and students), therefore, its generalizability is limited. However, the potential positive effects of SOP on work addiction have been confirmed in several other studies, but the findings on OOP and SPP are mixed [33,34,35]. Therefore, further research is required to clarify the mechanisms of these factors. In most of these studies, homogenous and/or small samples were used [32,34], and in some cases, not all components of the perfectionism scale were administered [33,35].

Although there are several studies confirming the relationship between low level of self-esteem and work addiction, the results are mixed [23,36,37,38,39]. Even though the correlations in these studies were all weak or moderate, a meta-analytic study by Clark and colleagues [10] verified the significant negative association between self-esteem and work addiction. It seems logical that perfectionism could decrease the level of self-esteem [40,41]. If the individual has to meet high standards set by significant others, then these standards will be internalized, and the individual will always want to be in accordance with these expectations. Another important attribute of such individuals is that their self-esteem is based on these perfectionist standards [42]. Therefore, it is assumed that a high level of perfectionism may increase the risk of work addiction via a lower level of self-esteem.

In summary, empirical evidence suggests that a reduced level of global self-esteem and elevated perfectionist tendencies are significant personality correlates of work addiction, but the mechanisms underlying these relationships are still unclear. Since cognitive factors may play an important role in work addiction [22,23], exploring cognitive mechanisms would help to better understand work addiction. For instance, rumination is one of the under-researched topics in work addiction, although it may explain specific thinking processes of individuals with work addictions. It has previously been reported that a low level of self-esteem can increase rumination [43]. On one hand, individuals with low self-esteem appear to experience more negative emotions when thinking about themselves [44]. On the other hand, individuals characterized by low self-esteem have a tendency to hide their feelings, problems, and/or personal failures from others [45], but nondisclosure of undesired feelings enhances ruminative thinking [46]. These reasons indicate that the lower the level of self-esteem, the more frequent individuals would experience rumination. It is assumed that this cognitive process could be a mediator between self-esteem and work addiction.

### 1.3. Rumination and Work Addiction

Rumination is a repetitive thinking process, and it has been defined by Nolen-Hoeksema as “repetitively focusing on the fact that one is depressed; on one’s symptoms of depression; and on the causes, meanings, and consequences of depressive symptoms” [47] (p. 569). Treynor, Gonzales, and Nolen-Hoeksema [48] differentiated between the ‘reflective pondering’ and ‘brooding’ factors of rumination. Brooding is a more maladaptive aspect of rumination, which reflects an inactive evaluation of an individual’s current unpleasant conditions [48]. However, reflective pondering is a more adaptive rumination process, described as a deliberate response to cognitive problem-solving and is less associated with depression than brooding [48].

It is well documented that individuals with work addiction experience several negative feelings (e.g., anxiety, guilt) when they are not working [11,49] and have a “high” or “rush” (positive emotions) during work [4]. Accordingly, work may act as a mood-modifying behavior or emotion regulation strategy for these individuals, which provides a distraction or an escape from negative emotions. It is worth noting that reaction to negative affects is considered to be more important in psychopathologies characterized by maladaptive behaviors than the elevated level of negative emotions itself. For instance, rumination on negative feelings, their causes and possible consequences [47] can easily lead to amplified negative emotions [50]. In addition, self-critical thoughts during rumination put an extra burden on the individual, therefore, it has been hypothesized that rumination is a potent risk factor in inducing maladaptive behaviors that provide engagement from ruminative thoughts and concomitant distress-related feelings [51]. Based on this finding, it is plausible to hypothesize that an increased tendency to ruminate is also a risk factor for work addiction. However, there is little empirical research supporting this hypothesis. In a Polish study, Wojdylo et al. [23] demonstrated that rumination had a positive moderate relationship with both work craving and work addiction. Interestingly, the authors theorized that rumination is a consequence of work craving and work addiction, and confirmed their hypotheses (i.e., both constructs predicted the level of rumination). Nevertheless, it is also reasonable to hypothesize that rumination in general is a risk factor for work addiction (i.e., the tendency for perseverant thinking about negative emotional states, their causes, and consequences to increase the risk of work addiction). To the present authors’ knowledge, this direction between work addiction and rumination has never previously been empirically examined.

Consequently, it is hypothesized that rumination not only predicts the risk of work addiction but also mediates the associations between both self-esteem and work addiction, and perfectionism and work addiction. Individuals who are described as having continuing ruminative thinking have a tendency to focus on and think about their failures and negative evaluations by others. They are often preoccupied with their mistakes and disappointments, which suggests that they are not satisfied with themselves, so they are not perfect. There have been empirical studies with both adults and adolescents that have verified the association between higher level of perfectionism and elevated rumination [52,53,54,55]. These findings were also statistically significant for both socially prescribed and self-oriented perfectionism. Based on the aforementioned theories and research, the following theoretical model was developed (see Figure 1):
a.Perfectionist individuals may be very sensitive about negative feedback and the results of their performance, which may determine their self-worth [40,41]. Such individuals always want to meet perfectionistic standards, and their failures result in lower self-esteem [56]. Therefore, a higher level of self-oriented and socially prescribed perfectionism predicts a lower level of self-esteem. Based on previous findings between self-esteem and work addiction, it is hypothesized that self-esteem mediates the relationship between both self-oriented perfectionism and work addiction, and between socially prescribed perfectionism and work addiction.b.Based on theories of the emotional processes of individuals with work addiction [4,49] and research examining the relationships between other behavioral addictions and rumination [57,58], it is hypothesized that a higher level of maladaptive rumination (i.e., brooding) predicts a higher level of work addiction.c.According to empirical evidence examining the relationship between low self-esteem and more intensive maladaptive rumination [59,60], it is hypothesized that brooding will mediate the relationship between self-esteem and work addiction.d.Since perfectionist individuals have high and unachieved standards and tend to more frequently ruminate about their failures and its negative aspects [52,53], it is hypothesized that maladaptive rumination (i.e., brooding) will mediate the relationship between both self-oriented perfectionism and work addiction, and socially prescribed perfectionism, and work addiction.

## 2. Materials and Methods

### 2.1. Participants and Procedure

The study was performed utilizing an online survey during April 2018 in Hungary. Before starting the survey that was shared by two of the largest national news portals, participants who were 18 years of age and older provided their consent to participate in the study. The other criterion for participation was having a current job at the time of data collection. Before completing the questionnaire, all participants could read information on the general aims of the study and the approximate duration of completing the survey. They were guaranteed confidentiality and anonymity, and their permission was achieved by choosing an option that they complied to participate in the study. No personal information was asked or saved. A total of 8511 participants began the survey, with 1929 individuals declining to take part in the study. A total of 2242 participants were eliminated from further analysis because they only completed the first pages (less than 10% of the online survey). A total of 4340 participants (female = 2202 [50.7%], male = 2138 [49.3%]) aged between 18 and 82 years (M = 37.4 years, SD = 9.9) were included in the final dataset. The present study was approved by the institutional review board (IRB) of the research team’s university and the study followed the guidelines of the Declaration of Helsinki.

### 2.2. Measures

#### 2.2.1. Sociodemographic Variables

The survey included questions concerning the main sociodemographic variables such as age, gender, level of education, and marital status.

#### 2.2.2. Self-Esteem

Self-esteem was assessed using the Rosenberg Self-Esteem Scale (RSES), which consists of 10 items [61,62]. The RSES is a self-report scale where participants respond by using a four-point Likert scale from “strongly agree” to “strongly disagree” (range from 10 to 40). Five reversed items (2, 5, 6, 8, 9) in this unidimensional scale showed a good internal consistency in the current sample (α = 0.878). Example item: “I am able to do things as well as most other people.”

#### 2.2.3. Perfectionism

The short version of the Multidimensional Perfectionism Scale (MPS) [63,64] was used to assess the three forms of perfectionism. The scale contains 15 items and assesses self-oriented perfectionism (SOP, five items; e.g., “One of my goals is to be perfect in everything I do”), other-oriented perfectionism (OOP, five items; e.g., “It does not matter to me when a close friend does not try their hardest”), and socially prescribed perfectionism (SPP, five items; e.g., “I feel that people are too demanding of me”). All the items on the OOP subscale were reversed. Individuals are asked to rate themselves on a seven-point Likert scale from “strongly agree” to “strongly disagree”. All three scales ranged from 5 to 35, and all showed good reliability in the current sample (SOP α = 0.844; OOP α = 0.790; SPP α = 0.808).

#### 2.2.4. Rumination

The short version of the Ruminative Response Scale [48] was used to assess rumination. The scale comprises 10 items and assesses two forms of rumination: brooding (with five items, e.g., “think ‘Why can’t I handle things better?’”) and reflective pondering (with five items, e.g., “Go away by yourself and think about why you feel this way”). The participants were asked to rate themselves on a four-point Likert scale from “almost never” to “almost always” to a question “How often do you…”. Both subscales ranged from 5 to 20 and had good reliability in the current sample (Brooding α = 0.742; Reflective pondering α = 0.743).

#### 2.2.5. Work Addiction

The risk of work addiction was assessed using the Work Addiction Risk Test Revised [7]. The original 25-item scale was developed by Robinson, Khakee, and Post [65], but the present study utilized the 17-item shortened version because it has better psychometric properties [7]. Individuals rate themselves on a four-point Likert scale from “never” to “always”. The higher the score on the scale, the higher the risk of work addiction (range from 14 to 68 points). The scale has good internal consistency in this study (α = 0.845). Example item: “I find myself continuing to work after my coworkers have called it quits.” All measures not already available in Hungarian were adapted using a translation/back-translation procedure [66].

### 2.3. Statistical Analyses

Descriptive statistics were performed using IBM SPSS statistics (Version 26) software (IBM Corp., Amonk, NY, USA) [67]. Before conducting path analysis, correlations between the variables included in the analysis were investigated. In cases it was relevant, the effect sizes of the correlations were statistically compared. After investigating the correlation matrix, a path analysis with MPlus 8.1 (Muthén & Muthén, Los Angeles, CA, USA) [68] was performed. There is agreement that robust maximum likelihood (MLR) estimation with bootstrap sampling provides better estimates with regard to standard errors compared to the maximum likelihood (ML) method, which is robust to deviation from normal distribution [69]. Therefore, because some of the variables examined in the present study were not normally distributed, we used the MLR estimator in the path analyses. To determine the significance of the indirect effects and the bias-corrected percentile confidence intervals (CIs) at 95%, standard errors and CIs were estimated with 1000 bootstrap samples. Although significance tests are helpful to decide which mediation effects are different from zero, the magnitude of the mediation effects are estimated with the proportion of the mediated effects in the total effects. In the case of samples including more than 500 participants, this procedure provides a reliable estimation of effect size of the mediation [70].

Multiple commonly used goodness of fit indices were used to evaluate the model fit: the chi-square goodness-of-fit statistic (χ^2^), the Tucker-Lewis fit index (TLI), the comparative fit index (CFI; ≥0.90 acceptable; ≥0.95 good), the root mean square error approximation (RMSEA; ≤0.08 adequate; ≤0.06 good) with its 90% CI, and the standardized root mean square residuals (SRMR; ≤0.08 adequate; ≤0.05 good) [71,72,73]. In the case of good fitting models, the chi-square test should not be significant (*p* > 0.05). However, a significant chi-square statistic is acceptable in large samples.

## 3. Results

### 3.1. Sample Characteristics and Correlations among Study Variables

Over half of the participants stated their place of residence as the capital city (60.2%), 15.2% of them lived in county towns, 17.5% lived in towns, and 7.2% reported living in villages. Regarding the highest level of education, most of the participants reported a level of education higher than secondary level (75.7%), 21.0% reported a secondary level, and only 2.2% reported a primary level of education. Three-quarters of the participants reported that they generally worked more than 40 hours per week and that the average hours they worked per week was 44.2 hours.

Zero-order correlations among the study variables are presented in Table 1. Significant positive and moderate associations were found between self-oriented perfectionism and other-oriented perfectionism, and between self-oriented perfectionism and socially prescribed perfectionism. However, the correlation between other-oriented perfectionism and socially prescribed perfectionism was very low, suggesting that they cover different aspects of perfectionism. While self-esteem showed only negligible correlations with both self-oriented and other-oriented perfectionisms, the association between self-esteem and socially prescribed perfectionism was significantly stronger. Both self-oriented and socially prescribed perfectionism correlated positively with brooding. However, the strength of the association was significantly stronger in the case of the socially prescribed perfectionism (z = 9.71, *p* < 0.001). Compared to brooding, the effect size of correlations between reflective pondering and the two perfectionism factors were much smaller. The correlation between socially prescribed perfectionism was again significantly stronger (z = 2.38, *p* < 0.01).

Self-esteem was negatively correlated with both brooding and reflective pondering. However, the association was significantly stronger with brooding (z = 9.90, *p* < 0.001). The risk of work addiction correlated positively with brooding and reflective pondering. Here again, the association between the risk of work addiction and brooding was significantly stronger (z = 3.59, *p* < 0.001). Finally, the risk of work addiction showed significant positive and moderate correlations with self-oriented and socially prescribed perfectionism, while the correlation with other-oriented perfectionism was significantly weaker (z = 5.41, *p* < 0.001; z = 4.29, *p* < 0.001, respectively).

### 3.2. Mediation Analysis

It was assumed that self-oriented perfectionism, socially prescribed perfectionism, and self-esteem would have both a direct and indirect effect on the risk of work addiction via the mediating effect of brooding as a maladaptive rumination. Other-oriented perfectionism was not used in the mediation analysis because it did not show significant correlations with brooding and reflective pondering, therefore, statistical mediation was not meaningful. The proposed mediation model (Figure 1) was tested with structural equation modeling (SEM). All the predictor and moderator variables were used as continuous observed variables. The model was fully saturated; therefore, the fit indices were not informative. However, trimming the non-significant paths resulted in the final model demonstrating an excellent degree of fit (χ^2^ = 16.9; df = 2; *p* < *0*.05; CFI = 0.997; TLI = 0.976; RMSEA = 0.041, 90% CI 0.025–0.061; SRMR = 0.008). The final model with standardized path coefficients is depicted in Figure 2.

The effect sizes of the proposed mechanisms were estimated and are reported in Table 2. The large proportion of mediation between socially prescribed perfectionism and the risk of work addiction was explained in the proposed model. The two paths involving brooding explained 44%, and the two paths involving reflective pondering explained 9% of the indirect effect. In the case of the self-oriented and other-oriented perfectionism factors, the explained mediated or indirect effects were negligible. The effect sizes of the indirect effects were also estimated separately between males and females (see Table 2) and received very similar patterns. Therefore, gender did not moderate the mediations.

## 4. Discussion

During the past two decades, a growing body of research has investigated the possible antecedents and consequences of work addiction. This significant increase in academic interest can be explained by (i) the increasing prevalence of work addiction in different countries [6,8,9]; and (ii) its several adverse physical and psychological correlates [4,10]. The individual risk factors of work addiction such as personality traits only explain a small amount of variance concerning work addiction [10,14]. However, low self-esteem and high perfectionism have repeatedly been found as important personality correlates of work addiction [32,37] but it was still unclear how these characteristics predicted work addiction. The present study found that maladaptive rumination (i.e., brooding) was an important mediator between self-esteem, perfectionism, and work addiction.

According to the correlation analysis and SEM, the results showed that the higher the level of brooding, the higher the risk for work addiction. Therefore, perseverative thinking (including focusing on the negative feelings and the negative aspects of the situations) was related to a higher risk of maladaptive behaviors such as work addiction. Consistent with this idea, Wojdylo et al. [74] found that those individuals who had better self-relaxation competencies reported a lower level of work craving than those individuals who were not as successful in self-relaxation. The results of our study indicate that rumination as a putatively maladaptive emotion regulation strategy appears to be associated with work addiction. At the same time, there is evidence for deficiency in emotion regulation and emotional intelligence in different forms of behavioral addictions such as online gaming disorder [75], gambling disorder [76,77], Internet addiction [78], and compulsive buying disorder [79,80]. If work addiction can be conceptualized as a behavioral addiction [81,82], similarities in emotion regulation processes between work addiction and other behavioral addictions can be presumed. The results here support this notion that there are similarities in underlying psychological mechanisms of well-known behavioral addictions (i.e., gambling and gaming disorders), and other, less investigated behavioral addictions (i.e., work addiction).

Second, it was hypothesized that self-esteem would mediate the relationship between both self-oriented perfectionism and work addiction, and between socially prescribed perfectionism and work addiction. Findings showed that all the forms of perfectionism had significant positive relationships with work addiction. However, while other-oriented perfectionism had only a weak positive association with work addiction, self-oriented and socially prescribed forms of perfectionisms were moderately related to work addiction. It is important that socially prescribed perfectionism appeared to have the most important role underlying work addiction. It predicted work addiction both directly and—via self-esteem—indirectly. Although the present cross-sectional study is not suitable for inferring causality, it can be posited that high standards derived from the social environment (e.g., from the parents, partner, or boss) may enhance the motivation for better achievement and excessive work. Alternatively, individuals with high socially prescribed perfectionism want to demonstrate their abilities, competence, and/or importance in work in order to gain more appreciation from important others.

The findings on perfectionism in the present study are in line with studies examining the underlying motivation of work addiction [83,84], demonstrating that work-addicted individuals are more characterized by introjected and extrinsic work motivations and in parallel they are not driven by intrinsic work motivations. For instance, these individuals want to perform better to avoid negative emotional states (e.g., shame, guilt and/or anxiety) or to gain more positive reinforcements from important others. Based on the findings here, it is plausible to hypothesize that these motivations are more prevalent among individuals with low self-esteem. The present study confirmed that a higher level of socially prescribed perfectionism had an association with lower self-esteem, which then related to increased work addiction tendencies. It appears logical to suggest that the self-esteem of such individuals is based on conditions, the perfectionist standards [42]. Working excessively and obsessively are possibilities for individuals to both accept themselves better and meet the perfectionist standards of others. Consequently, in future studies, the role of contingent self-esteem [85] in work addiction should be further investigated.

It appears reasonable to assume that individuals who have low self-esteem, repeatedly and persistently think about their failures in different areas of their life including work. They may also ruminate about their negative emotional states, mistakes, difficulties, and/or imperfections at work. Therefore, they may feel the desire to work more and try to perform better and better. Consequently, it was assumed that brooding would mediate the relationship between self-esteem and work addiction. Furthermore, it could be argued that rumination can decrease self-esteem in the long term. However, prospective studies testing any cause-and-effect relationships between self-esteem and rumination are limited. In a longitudinal study, Kuster, Orth, and Meier [43] conducted mediation analysis and found that low self-esteem predicted following rumination in an eight-month period. The results here are in line with Kuster et al.’s [43] study, and maladaptive rumination was predicted by a lower level of self-esteem. As the present study found, rumination also mediated between lower self-esteem and higher risk of work addiction. This result indicates that future studies are needed to explore the emotion regulation strategies in work addiction.

Finally, it was hypothesized that perfectionist individuals are more prone to ruminate about their failures and its negative aspects more frequently, based on previous work [53,54] and that maladaptive rumination would mediate the relationship between perfectionism and work addiction. The results supported this hypothesis. Perfectionism as a personality risk factor for work addiction has been repeatedly confirmed [10], although the possible underlying cognitive mechanisms are still unclear. The present study’s results suggest that the symptoms of work addiction could escalate if a perfectionist person ruminates about their negative emotional states and failures. It is especially true for those individuals who want to be perfectly prescribed by others and who want to demonstrate their suitability or competence to their parents, family members, friends and/or colleagues.

Although Wojdylo et al. [23] reported a positive relationship between rumination and work addiction, and they tested the effects of work addiction on ruminative processes, as far as we know, this study is the first to test the explanatory value of maladaptive rumination in work addiction. The novelty of the study is demonstrating that brooding type of rumination as a putatively maladaptive strategy explains why individuals characterized by low self-esteem and high perfectionism have a higher risk of work addiction. Rumination is frequently a normal response especially immediately after the negative event. However, more adverse consequences (such as work addiction) are found if a stable ruminative response style meets with a higher level of perfectionism. Therefore, the present study helps us to better understand the individual factors underlying work addiction. At the same time, it should be recognized that both directions between self-esteem, perfectionism, and rumination are plausible and only prospective studies can identify the relevance of these casual relationships.

The present study indicates that cognitive mechanisms and emotion regulation processes are important factors in work addiction. Previous studies have highlighted the deficits in emotional processes between different addictive disorders: alexithymia [86], a lower level of emotional intelligence [87], difficulties in emotional regulation [88], or more frequent ruminative processes [89]. The novelty and the utility of the present study is that it implies similar cognitive-affective mechanisms in work addiction as has been found in other addictive disorders (i.e., gambling disorder, gaming disorder, problematic Internet use, and compulsive buying disorder). Since there is still debate on work addiction concerning its possible status among behavioral addictions [1,90,91], the findings here may contribute to the discussion by highlighting these parallels. Future studies on both cognitive and emotional processes of work addiction may better help clarify the position of work addiction among behavioral addictions more generally.

Additional suggestions for future studies would be to explore the relationship between work addiction and ‘work-related rumination’. Individuals commonly think about work issues during leisure time such as unfinished tasks, important projects, future presentations or meetings, and/or critiques by others (e.g., line managers or colleagues). However, some individuals are characterized by ‘work-related rumination’, perseverant thinking about work during free time or vacation [92], and ‘unwinding processes’, namely inadequate return or poor disconnection from work [75]. Cropley and colleagues [93] developed a psychometric instrument for assessing work-related rumination (three factors comprising ‘distraction’, ‘problem-solving pondering’, and ‘affective rumination’) and found that a higher level of work-related rumination had an association with more frequent cognitive failures and deficits in several key skills for productive work [94]. In addition, work-related rumination has been found to correlate with negative health outcomes [95,96], sleep disturbances [97], and unhealthy eating habits [93]. However, to the best of our knowledge, the associations between work-related rumination and work addiction have never previously been investigated. The present study focused on general ruminative processes because several theoretical models have suggested perseverative thinking patterns among individuals with work addiction [18,19,20,21], but in the future, it would be important to investigate the role of different possible forms of work-related rumination in work addiction.

Although the present study broadens the understanding of the individual risk factors of work addiction, it is not without limitations. First, the study was cross-sectional, therefore it was not possible to determine any cause-and-effect relationships. Second, the study comprised a convenience sample of adult workers. Although the sample was large, diverse, and the gender ratio was good, it was not representative, which narrows the generalization of the findings. In the future, representative samples should be utilized. Third, the sample was heterogeneous according to the workplaces, professions, and occupations, but these variables were not included in the analysis as control variables, limiting the generalizability of the findings. Future studies should also include these control factors in the analyses. Fourth, self-report questionnaires were utilized for assessing personality factors, ruminative responses, and the level of work addiction. Other important biases may have occurred because of the self-reported nature of the research (e.g., socially desirable responses or memory recall biases). At the same time, the level of work addiction may be explored better by using collateral (e.g., spouse, colleague) ratings or 360° employee ratings of individuals rather than relying on one source of data [98]. Fifth, the Work Addiction Risk Test Revised is only a screening measure for estimating the risk of work addiction, but it is not a diagnostic tool. Sixth, among cognitive-emotional processes, the study focused only on rumination. However, other maladaptive strategies such as catastrophizing or depressive tendencies may also be important contributors in the development of work addiction. A limitation of the study that these other possible factors were not investigated or controlled for, and future studies should also examine these other aspects. Finally, although the development and maintenance of work addiction is a complex phenomenon comprising individual, social, societal, and cultural factors [13,98], the present study did not control for these meso- and macro-level social variables.

## 5. Conclusions

Work addiction has been conceptualized as a problem strongly related to personality factors and several studies have demonstrated that lower levels of self-esteem and higher levels of perfectionism are important correlates of work addiction. In the present study, the role of rumination in the relationships between personality factors and work addiction was explored and the results suggested that brooding, as a maladaptive ruminative response, might play an important role in the cognitive mechanisms underlying work addiction. In conclusion, individuals at risk of work addiction tend to ruminate more frequently and this type of thinking process mediates between perfectionism (especially socially prescribed perfectionism) and work addiction, and between low self-esteem and perfectionism. It is recommended that future studies should explore cognitive mechanisms in work addiction more exhaustively because these types of studies may help to better understand the underlying mechanisms of work addiction.

## Figures and Tables

**Figure 1 ijerph-17-07332-f001:**
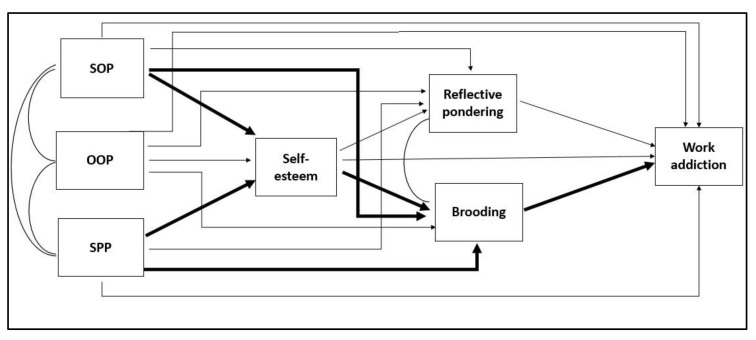
A theoretical model of the mediating effect of brooding between (i) self-esteem and work addiction, (ii) self-oriented perfectionism and work addiction, and (iii) socially prescribed perfectionism and work addiction. Notes: The arrows representing the study’s hypotheses are emboldened. SOP = self-oriented perfectionism, OOP = other-oriented perfectionism, and SPP = socially prescribed perfectionism.

**Figure 2 ijerph-17-07332-f002:**
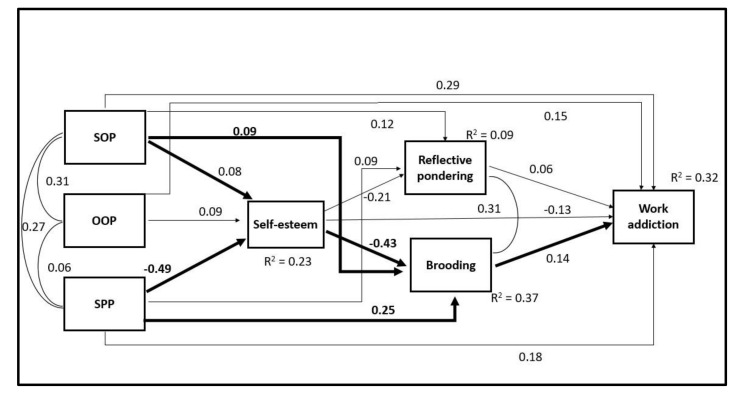
The final model with standardized path coefficients. Notes: All estimations were significant at the *p* < 0.001 level. The arrows representing the study’s hypotheses are emboldened. SOP = self-oriented perfectionism, OOP = other-oriented perfectionism, and SPP = socially prescribed perfectionism.

**Table 1 ijerph-17-07332-t001:** Means, standard deviations, reliabilities, and inter-correlations of self-esteem, self-oriented perfectionism, other-oriented perfectionism, brooding, reflective pondering, and work addiction.

	1.	2.	3.	4.	5.	6.	7.
1. Work addiction	(0.84)						
2. Self-esteem	***−0.28***	(0.88)					
3. Brooding	***0.35***	***−0.54***	(0.74)				
4. Reflective pondering	***0.22***	***−0.26***	***0.41***	(0.74)			
5. Self-oriented perfectionism	***0.41***	−0.02	***0.17***	***0.15***	(0.84)		
6. Other-oriented perfectionism	***0.24***	***0.09***	0.02	−0.02	***0.31***	(0.79)	
7. Socially prescribed perfectionism	***0.39***	***−0.46***	***0.47***	***0.22***	***0.27***	**0.06**	(0.81)
Range	17–68	10–40	5–20	5–20	5–35	5–35	5–35
Mean	42.41	28.61	10.32	10.74	26.51	20.35	17.08
Standard deviation	7.64	5.41	3.09	3.09	5.84	6.05	6.95

Note: Pearson correlations, *N* = 4340. Emboldened correlations are significant at least at *p* < *0*.05. Italicized correlations are significant after Bonferroni correction (the corrected *p*-value <0.0024). Cronbach’s α coefficients are reported on the diagonal in parentheses. The range of covariance coverages was 0.997–1.000.

**Table 2 ijerph-17-07332-t002:** Standardized estimates of the direct and total indirect effects on the risk of work addiction and mediator variables and their respective confidence intervals.

Path	Total Effect [95% CI]	Indirect Effect [95% CI]	Proportion of Total Effect
Total sample *N* = 4340			
Self-oriented perfectionism → Work addiction	0.289 [0.261–0.317]	0.004 ^#^ [−0.004–0.012]	N/A
Socially prescribed perfectionism → Work addiction	0.306 [0.279–0.332]	0.130 [0.112–0.148]	42.5% *
Specific paths			
Socially prescribed perfectionism → Brooding → Work addiction		0.031 [0.022–0.040]	10.1%
Socially prescribed perfectionism → Reflective pondering → Work addiction		0.006 [0.003–0.010]	2.0%
Socially prescribed perfectionism → Self-esteem→ Brooding → Work addiction		0.026 [0.019–0.034]	8.5%
Socially prescribed perfectionism → Self-esteem→ Reflective pondering → Work addiction		0.006 [0.003–0.010]	2.0%
Males *N* = 2138			
Self-oriented perfectionism → Work addiction	0.262 [0.221–0.303]	0.002 ^#^ [−0.010–0.016]	*n*/A
Socially prescribed perfectionism → Work addiction	0.315 [0.277–0.352]	0.142 [0.116–0.169]	45.1% *
Specific paths			
Socially prescribed perfectionism → Brooding → Work addiction		0.036 [0.023–0.052]	11.4%
Socially prescribed perfectionism → Reflective pondering → Work addiction		0.005 [0.001–0.011]	1.6%
Socially prescribed perfectionism → Self-esteem→ Brooding → Work addiction		0.031 [0.020–0.043]	9.8%
Socially prescribed perfectionism → Self-esteem→ Reflective pondering → Work addiction		0.007 [0.002–0.012]	2.2%
Females *N* = 2202			
Self-oriented perfectionism → Work addiction	0.316 [0.278–0.353]	0.003 ^#^ [−0.008–0.014]	N/A
Socially prescribed perfectionism → Work addiction	0.289 [0.250–0.326]	0.115 [0.091–0.139]	39.8% *
Specific paths			
Socially prescribed perfectionism → Brooding → Work addiction		0.025 [0.014–0.038]	8.7%
Socially prescribed perfectionism → Reflective pondering → Work addiction		0.006 [0.002–0.012]	2.1%
Socially prescribed perfectionism → Self-esteem→ Brooding → Work addiction		0.021 [0.012–0.031]	7.3%
Socially prescribed perfectionism → Self-esteem→ Reflective pondering → Work addiction		0.006 [0.002–0.010]	2.1%

Note: Standardized effects are reported. [95% CI]: Biased corrected bootstrapped confidence interval. * The specific paths were estimated when the explained proportion of association was relevant (i.e., larger than 25%). ^#^ The sum of the absolute values of indirect paths were 0.023, 8% in the total sample, 0.026, 8.2% in females, and 0.050, 19.0% in males. NS = non-significant indirect effect.

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
