# Peer review of "Maladaptive Rumination Mediates the Relationship between Self-Esteem, Perfectionism, and Work Addiction: A Largescale Survey Study"

_ijerph, 2020, doi:10.3390/ijerph17197332_

Round 1
Reviewer 1 Report
Final Verdict Manuscript ID ijerph-913108
The research work titled “Maladaptive rumination mediates the relationship between self - esteem, perfectionism, and work addiction: A largescale survey study” of Bernadette Kun and collaborators.
In this study, tools four were used: Work Addiction Risk Test Revised, Rosenberg Self- Esteem Scale, Multidimensional Perfectionism Scale, and Ruminative Response Scale. In order to better understand work addiction.
It is an excellent investigation it is suggested to correct some specific points:
Line 4. Remove the words "large scale". Since the sample was large, but not representative, which reduces the generalizability of the findings.
Line 278. What software did you use for the statistical analysis of your data?
Line 488. Your conclusions are very bad. In addition, due to the type of research they carried out, no references need to be added in conclusions, please delete them.
Author Response
- Line 4. Remove the words "large scale". Since the sample was large, but not representative, which reduces the generalizability of the findings.
- Thank you for your comment. Although our study was not representative, it does not mean that it was not largescale because we have involved more than 4,300 individuals in our analysis. We would be happy to keep this title to emphasize the expansive size of our sample. However, for accentuating that our study was unrepresentative, we add this word in the abstract (see page 1, line 25.). We hope that you can accept our opinion.
- Line 278. What software did you use for the statistical analysis of your data?
- Thank you for your feedback. We have added a sentence on page 6, line 264, namely: “Descriptive statistics were performed using IBM SPSS statistics (Version 26) software [98].” However, in the first version of the manuscript, we already reported the other software we used, see page 6, lines 266-267. “After investigating the correlation matrix, a path analysis with MPlus 8.1 [65] was performed.”
- Line 488. Your conclusions are very bad. In addition, due to the type of research they carried out, no references need to be added in conclusions, please delete them.
- Thank you for your feedback. We have rewritten our conclusions (see page 13, line 491-501.)
Reviewer 2 Report
First of all, there is no need to list too many references, as an article in a journal, just list the articles that are cited in the text.
Secondly, the second part (methodology) and the final conclusion are too short, especially the conclusion, which should be extended.
After all, this article is a result of an extensive analysis of experimental data and deserves further refinement.
Author Response
- First of all, there is no need to list too many references, as an article in a journal, just list the articles that are cited in the text.
- This comment suggests that we listed some extra references beyond the papers we have cited in the text. We checked all the citations and references in the manuscript again, and we have not found any unnecessary or irrelevant references in our list (i.e., all the papers in the ‘References’ list were cited in the text of our paper).
- Secondly, the second part (methodology) and the final conclusion are too short, especially the conclusion, which should be extended. After all, this article is a result of an extensive analysis of experimental data and deserves further refinement.
- Thank you for your feedback. We have now rewritten our conclusions (see page 13, line 491-501.).
- Regarding the methodology part of the manuscript, in the first version of the manuscript, we presented the characteristics of the sample (participants), the process of the procedure, all the measures we used, and the statistical analyses we performed. We think that the most important information can be found in the Methods section of the paper. If there is something missing in our methodology, please specify them, and we will be happy to add such information. We would also like to point out that one of the other reviewers suggested that one part of ‘Participants and procedure’ (i.e., characteristics of the sample) should be moved from the ‘Methods’ to the ‘Result’ section of the manuscript. We have carried out the other reviewer’s request, therefore, the following section has been moved to the ‘Results’ section (page 7, lines 287-292): “Over half of the participants stated their place of residence as the capital city (60.2%), 15.2% of them lived county towns, 17.5% lived in towns, and 7.2% reported living in villages. Regarding the highest level of education, most of the participants reported a level of education higher than secondary level (75.7%), 21.0% reported secondary level, and only 2.2% reported a primary level of education. Three-quarters of the participants reported that they generally worked more than 40 hours per week and that the average hours they worked per week was 44.2 hours.”
Reviewer 3 Report
Interesting paper but I find it difficult to follow with too much information that is poorly organized. Did not address the role of culture and rumination as well as work addiction. The main weakness of the study are those delineated by authors in last paragraph. The body of the manuscripts and mainly the discussion section is more definitive than the methodology warrants. As it was pointed out, this is cross-sectional study and unable to determine a cause and effect. The discussion suggests cause and effect This is not warranted. The findings are very specific to the undifferentiated population studied and can't be generalized beyond that. As such, the relevance of the interesting findings are quite limited.
Author Response
Interesting paper but I find it difficult to follow with too much information that is poorly organized.
- Did not address the role of culture and rumination as well as work addiction. The main weakness of the study are those delineated by authors in last paragraph.
- Thank you for your comment about a very important topic, namely, the role of culture in work addiction. In our study, we only focused on the individual (personality) correlates of work addiction. However, we are fully aware of the importance of social, societal, and cultural factors. Involving these other meso- and macro-level factors would have extended the analyses and the size of the manuscript significantly. Therefore, we did not to include these variables in our analyses. Nevertheless, we have added an extra paragraph to the ‘Introduction’ (page 2, lines 52-61) and in the ‘Discussion’, and we have marked this issue as another limitation of our study (page 12, lines 485-487).
- The body of the manuscripts and mainly the discussion section is more definitive than the methodology warrants. As it was pointed out, this is cross-sectional study and unable to determine a cause and effect. The discussion suggests cause and effect. This is not warranted.
- Thank you for your comment. We absolutely agree with your observation. Therefore, we have made several changes in the phrasing of the explanation of our results (see page 10, lines 361, 366-367, page 11, lines 386-387, line 435). Moreover, in our revised manuscript, we have emphasized that the design of our study is not suitable for cause and effect relationships (see page 11, lines 386-387; page 12, lines 467-468).
- The findings are very specific to the undifferentiated population studied and can't be generalized beyond that. As such, the relevance of the interesting findings are quite limited.
- Thank you for this comment. In the ‘Discussion’ (page 12, lines 468-471), among the limitations of our study, we already pointed out that “…the study comprised a convenience sample of adult workers. Although the sample was large, diverse, and the gender ratio was good, it was not representative which narrows the generalization of the findings. In the future, representative samples should be utilized.” In the current, revised version, we have added two more sentences (page 12, lines 471-474) to underline the heterogeneous characteristics of our sample as you suggested.
Reviewer 4 Report
This is an interesting paper describing the relationship of rumination and self-esteem, perfectionism, and work addition. My comments are follows. Hope it helps to improve this paper.
The introduction is definitely too long. Authors should clearly define the introduction to the research topic. Some of the explanations contained in the introduction (i.e. theoretical models) can be transferred to the discussion. There is no one clear definition of rumination and work addiction. Please show directly the epidemiological context in the problem.
Page 5 line 225-232. It should be remove from section of Materials and Methods because these are results of Yours study.
Page 7, line 320 Table 1 is unclear (unreadable). would need to be redrafted.
Conclusions. What kind of other addictive disorders are exactly similar to mechanism in work addiction?
Author Response
This is an interesting paper describing the relationship of rumination and self-esteem, perfectionism, and work addition. My comments are follows. Hope it helps to improve this paper.
- The introduction is definitely too long. Authors should clearly define the introduction to the research topic. Some of the explanations contained in the introduction (i.e. theoretical models) can be transferred to the discussion.
- Thank you for your comment. While we respect your opinion, we think that it is very important to help those readers of the International Journal of Environmental Research and Public Health who are not familiar with ‘work addiction’ to introduce this problem in a sufficient way. The length of our manuscript does not exceed the word limit. However, we consider your suggestion and we have deleted several sentences from the Introduction, the following parts:
Although several theories have emphasized the relevance of obsessive tendencies in work addiction [2,13,14,16], there is very limited empirical evidence of the association between these constructs. Mudrack [20] and Aziz and colleagues [21] both found that obsessive-compulsiveness showed a significant positive but weak correlation with the risk of work addiction. At the same time, it is also presumed that work addiction shows some overlap with obsessive-compulsive disorder (OCD) or, more specifically, obsessive-compulsive personality disorder (OCPD). Among the diagnostic criteria of OCPD [22], it is listed that the individual shows high perfectionism that conflicts with effective achievement, and that the extreme dedication to work and productivity means that they neglect social relationships, do not take rest, or engage in any hobbies. Based on these OCPD characteristics and the several common factors of both OCPD and behavioral addictions (e.g., gambling disorder) [23, 24], the high correlation between work addiction and OCPD is expected. Only two studies have investigated the association between work addiction and OCD [25] and OCPD [26]. While in a Norwegian study [25] only weak relationship was found between work addiction and OCD, in an American study [26] OCPD (assessed using the Schedule for Adaptive and Nonadaptive Personality, SNAP) [27] reported a strong positive correlation with work addiction (r=0.64, p<.001). This latter finding underlines the possible interference between work addiction and OCPD. However, it should be noted that the SNAP contains items that refer to work addiction symptoms, which means there is a specific artefact in this regard.
“In addition, in all the aforementioned studies assessing any components of the multidimensional perfectionism scale, the Dutch Work Addiction Scale was utilized and there are no data about other frequently used work addiction scales (e.g., Work Addiction Risk Test or Bergen Work Addiction Scale).”
- There is no one clear definition of rumination and work addiction.
- Thank you for your observation. We have now added the definitions of both work addiction (page 2, lines 69-72.) and rumination (page 4, lines 138-140.) in the revised version of our manuscript.
- Please show directly the epidemiological context in the problem.
- You are right to point out that is very important to show the epidemiological context of work addiction. In the first version of our manuscript, we clearly reported the existing data on the prevalence of work addiction (page 2, lines 46-51): “Although the prevalence of work addiction varies across surveys due to the different definitions, methods, and screening instruments used, largescale representative studies have shown that the lifetime prevalence of work addiction prevalence was 8.3% among Norwegian workers [6], and 8-9% among Hungarian employees [7, 8]. Due to its seemingly high prevalence compared to other addictive behaviors [9], there is an increasing interest in understanding the psychosocial correlates of work addiction [4, 10].” We are of the view that this information is sufficient for an epidemiological context, but if you think we have overlooked any key studies we will be happy to add them to our manuscript.
- Page 5 line 225-232. It should be remove from section of Materials and Methods because these are results of your study.
- Thank you for your comment. We have moved one part of the sample characteristics from the ‘Methods’ section part to the ‘Results’ section of the paper (see page 7, lines 287-292).
- Page 7, line 320 Table 1 is unclear (unreadable). Would need to be redrafted.
- If Table 1 is not readable, this would indeed be very problematic. However, we can read both the title and the table without any problem. Therefore, it is not clear for us, what type of problem have you detected. Is there a technical problem (i.e., the data of the table are not visible or merged)? Or do you mean that we should rephrase something (i.e., the title of the table is not exactly clear)? Please specify exactly what the problem is so that we can rectify this.
- What kind of other addictive disorders are exactly similar to mechanism in work addiction?
- Thank you for your comment. In the Discussion (page 10, lines 371-378), we specified those behavioral addictions (i.e. gaming disorder, gambling disorder, problematic internet use, and compulsive buying disorder) which can be characterized by similar emotional dysregulation processes. “At the same time, there is evidence for deficiency in emotion regulation and emotional intelligence in different forms of behavioral addictions, such as online gaming disorder [74], gambling disorder [72, 73], problematic internet use [75] and compulsive buying disorder [76, 77]. If work addiction can be conceptualized as a behavioral addiction [78, 79], similarities in emotion regulation processes between work addiction and other behavioral addictions can be presumed. The results here support this notion that there are similarities in underlying psychological mechanisms of well-known behavioral addictions (i.e., gambling and gaming disorders), and other, less investigated behavioral addictions (i.e., work addiction).” But you are right that in a later point of the ‘Discussion’, we did not specify these disorders, therefore, we have supplemented this part of the manuscript (page 12, lines 443-444).
Round 2
Reviewer 3 Report
The manuscript is significantly improved. Thank you for addressing my concerns and I am glad that my feedback helped you in producing an acceptable manuscript. Work addiction is an important topic that requires good research. The introduction is lengthy but I understand the need to go in detail on the subject presented. The information provided is better organized and you have provided adequate operational definitions of the subject matter as needed. The results section is much improved and easier to follow and understand. Glad to that you addressed the fact that a cause and effect could not be stablished given the nature of the research design. You have addressed the limitations of the study this fine adequately.